# Development of One-Step Non-Solvent Extraction and Sensitive UHPLC-MS/MS Method for Assessment of N-(n-Butyl) Thiophosphoric Triamide (NBPT) and N-(n-Butyl) Phosphoric Triamide (NBPTo) in Milk

**DOI:** 10.3390/molecules26102890

**Published:** 2021-05-13

**Authors:** Chikere G. Nkwonta, Macdara O’Neill, Niharika Rahman, Mary Moloney, Patrick J. Forrestal, Sean A. Hogan, Karl G. Richards, Enda Cummins, Martin Danaher

**Affiliations:** 1Food Safety Department, Teagasc Food Research Centre, Ashtown, D15KN3K Dublin, Ireland; mary.moloney@teagasc.ie (M.M.); martin.danaher@teagasc.ie (M.D.); 2Teagasc, Johnstown Castle, Co., Y35 Y521 Wexford, Ireland; macdara.oneill@teagasc.ie (M.O.); Niharika.Rahman@teagasc.ie (N.R.); Patrick.forrestal@teagasc.ie (P.J.F.); karl.richards@teagasc.ie (K.G.R.); 3Food Chemistry and Technology Department, Teagasc Food Research Centre, Moorepark, Fermoy, Co. Cork, P61 C996 Cork, Ireland; sean.a.hogan@teagasc.ie; 4School of Biosystems and Food Engineering, Agriculture and Food Science, University College Dublin, D04 V1W8 Belfield, Ireland; enda.cummins@ucd.ie

**Keywords:** UHPLC-MS/MS, urease inhibitor, ultrafiltration extraction, N-(n-butyl) thiophosphoric triamide, N-(n-butyl) phosphoric triamide, milk, residue, cattle, stability

## Abstract

N-(n-butyl) thiophosphoric triamide (NBPT) is a urease inhibitor utilised in urea-based fertilizers. In Ireland, fertilizer treated with NBPT is applied to pasture to mitigate both ammonia and nitrous oxide emissions, but concerns arise as to the potential for residues in milk products. A quick ultrafiltration extraction and ultra-high performance liquid chromatography coupled with mass spectrometry triple quadrupole (UHPLC-MS/MS) quantitation method was developed and validated in this study. The method was applied in the analysis of samples collected from a field study investigating potential transfer of NBPT residues into milk. NBPT and NBPTo residues, were extracted from fortified milk samples and analysed on a UHPLC-MS/MS with recoveries ranging from 74 to 114%. Validation of the UHPLC-MS/MS method at low (0.0020 mg kg^−1^) and high (0.0250 mg kg^−1^) concentration levels in line with SANTE/12682/2019 showed overall trueness in the range of 99 to 104% and precision between 1 and 10%, RSD for both compounds. The limit of quantitation (LOQ) was 0.0020 mg kg^−1^ and other tested parameters (linearity, sensitivity, specificity, matrix effect, robustness, etc.) satisfied acceptance criteria. Stability assessment using spiked samples revealed the compounds were stable in raw and pasteurised milk for 4 weeks at –80 °C storage temperature. Maintaining samples at pH 8.5–9.0 further improved stability. Analysis of 516 milk samples from the field study found that NBPT and NBPTo concentrations were below the LOQ of 0.0020 mg kg^−1^, thus suggesting very low risk of residues occurring in the milk. The method developed is quick, robust, and sensitive. The method is deemed fit-for-purpose for the simultaneous determination of NBPT and NBPTo in milk.

## 1. Introduction

Inhibitor technology, including urease and nitrification inhibitors, are useful tools for reducing environmental losses from nitrogen-based fertilisers. The urease inhibitor, N-(n-butyl) thiophosphoric triamide (NBPT), has proven to be a very effective tool in mitigating ammonia loss from urea [1,2]. In Ireland, the use of urea with NBPT is encouraged as a mitigation measure to reduce both ammonia and nitrous oxide emissions [2]. In 2005, it was reported that 54% of non-CO_2_ greenhouse gas emissions were produced by agriculture [3,4]. Studies have also shown that NBPT enables efficient utilisation of urea by delaying urea hydrolysis, making N available to crops/plants, thereby enhancing agricultural yield and profitability but reducing ammonia losses. For instance, higher maize yield (3.280–3.567 t ha^−1^) was observed in plots treated with varying amounts of NBPT-coated urea compared to plots treated with urea only (3.270 t ha^−1^) [5]. In addition, NBPT was reported to reduce the number of side-dress applications in maize farming as only a single application of NBPT-coated urea fertiliser gives same yield of maize crop as a double side-dress application of conventional urea fertiliser [6]. Globally there is concern over chemical residues in food products and an increasing need to ensure that chemicals used in agricultural production do not impact food safety. This was highlighted when residues of the nitrification inhibitor dicyandiamide was observed in milk products from New Zealand [7]. To examine this question, it is necessary to understand the compound and develop a sensitive method to detect NBPT/NBPTo. A highly sensitive method is needed, as NBPT is used at relatively low concentrations in fertilisers. The maximum permissible level for NBPT under EU fertiliser regulations is 0.2% by mass of N in urea fertilisers [8].

NBPT, with chemical formula C_4_H_14_N_3_PS (Figure 1), has been reported to degrade more rapidly in acidic to slightly alkaline environment (pH 5.1–7.6). It has a half-life of 0.07 days at pH 5.1, 0.59 days at pH 6.1, 2.7 days at pH 7.6, and 3.43 days at pH 8.2 [9,10,11,12]; however, it showed stability in milk samples fortified at concentrations of 1 mg kg^−1^ and stored at −20 and −80 °C [13]. These reports indicate that both pH and temperature conditions are important factors to consider when assessing its potential for transfer into milk and dairy food products. A toxicity assessment by the National Australian Industrial Chemicals Notification and Assessment (NICNAS) [14] characterised NBPT as having a significant negative impact on the reproductive system of male and female rats. Therefore, it is seen as a Category 3 substance capable of causing harm to fertility [14]. Furthermore, it showed neurotoxic and hepatotoxic effects on the animal test subjects at concentration levels of 200–5000 mg kg^−1^ bw^−1^. Direct feeding of NBPT, resulted in high body weight concentrations of NBPT in rats and indicate a need for a risk assessment to determine how much, if any, NBPT might bio-transfer from pasture use, through the grazing animal and into milk. 

A dosing study, which fed NBPT in capsule form directly to cows, as opposed to the pasture-applied fertiliser, has detailed a sample processing and UHPLC-MS/MS analytical method for simultaneous detection of NBPT and NBPTo in cow’s milk and other matrices (muscle, liver, kidney, etc.). The method achieved a LOQ of 0.05 mg kg^−1^ (milk matrix) [13]. Currently, there are no legislative requirements established on potential residue levels for NBPT and NBPTo in cow’s milk, and other food products. Hence, it is pertinent that field trials simulating real farm scenarios, are performed, and a database developed to that effect, as a basis for making informed decisions by stakeholders and policy makers in the near future. Additionally, other efficient sample separation and UHPLC-MS/MS analytical techniques achieving lower quantitation or reporting limits need to be developed as alternatives to existing ones, since lower concentrations are more likely to be found in milk and dairy products, if present. The present study focused on developing a one-step, sample separation technique, and a robust, sensitive LC-MS/MS analytical method aimed at achieving acceptable analyte recovery and low limit of quantitation (LOQ). Secondly, it aimed to develop a NBPT/NBPTo residue database from analysing samples (using the method developed) obtained from a dairy farm experiment using protected urea as the fertiliser N source (NBPT farm) and a commercial farm that was not using a protected urea (Negative Control farm).

## 2. Results and Discussion

### 2.1. Sample Extraction Development

A rapid ultracentrifugation sample separation method that involved placing 4 mL of milk sample in an ultrafiltration centrifuge (UFC) tube containing regenerated cellulose filter membrane (Ultracel^®^-3K, 3000 NMWL), and centrifuging at 4000 relative centrifugal force (rcf) for 15–30 min at 4 °C was applied. During method development, a range of different ultrafiltration filter membranes—polyethersulfone (PES), polypropylene and cellulose—with varying binding capacities and tube sizes, were trialed. A final selection was made based on the filter unit that allowed simultaneous passage of both NBPT and NBPTo residues with sufficient acceptable recovery. To address the issue of degradation during extraction, the concentration and minimum volume of buffer that would achieve the required pH level (pH 8.5–9.0) was determined. This volume (200 µL) was added to samples (4 mL) in the tube before centrifugation. Preliminary in-house experiments applying solvent (MeCN) extraction and QuEChERS-based methods was also performed. However, the methods were deemed unsuitable because they involved long periods of evaporation that promoted degradation of the target analytes (NBPT and NBPTo). Again, temperature increase of the QuEChERS salting-out reaction system affected analyte degradation, leading to low recovery results. The ultrafiltration technique with device filter units lined with different types of membranes has continued to gain ground in the analytical industry because of the simple and rapid nature of the procedure. A number of studies utilizing UFC filters for purification of different types of molecules have been reported [15,16,17]. The principle, which is based on centrifugation, forces the buffer containing free residue molecules through the size-exclusion membrane to achieve a fast separation free from protein bound residues [17]. The use of Ultra-4 Centrifugal filter tubes with low binding regenerated cellulose membranes allowed quick extraction of spiked NBPT and NBPTo concentrations from ammonium carbonate-buffered cow milk by ultrafiltration through the membranes, which had a molecular weight cut- off (MWCO) of 3000 k. In addition, the heat-sealed membranes minimized downstream extractables like proteins and fat molecules, thereby eliminating potential contaminants capable of causing interference in LC-MS/MS detection instruments. The low binding regenerated cellulose membrane also reduced the potential for non-specific binding of the residues, which could hinder its ultrafiltration [17]. Leaving the samples in the UFC tube to stand for 10 min at 3–4 °C before centrifugation was found to be very beneficial as the filters were well activated and potential for a cloudy filtrate was eliminated. Furthermore, it improved consistency and uniformity, for all samples, in the volume of filtrate produced at the end of centrifuge run.

The average extraction recovery, for both analytes, from duplicate milk samples spiked at low (0.0020 mg kg^−1^) and high (0.0250 kg^−1^) levels, in ten (10) different analytical runs were consistently between 74–114% with mean values for NBPT and NBPTo of 93 and 96% recovery, respectively. These values are within the acceptable range (70–120%) recommended by the European Commission regulation document SANTE/12682/2019 [18], thus, verifying that the extraction method was fit for purpose. The NBPT recovery result was the same as that obtained in a previous study, which reported overall per cent recovery for NBPT, extracted from fortified milk using a solvent extraction and centrifugation method, to be 93.2 ± 3.2%. The fact that the previous study achieved the recovery following fortification of milk samples at much higher levels (0.05, 1, and 10 mg kg^−1^) compared to that used in this study (0.002 and 0.0250 mg kg^−1^) suggests that the ultracentrifugation method developed and reported here is more efficient. 

### 2.2. UHPLC-MS/MS Method Development

The MS/MS optimization experiment using Teed-infusion showed that both NBPT and NBPTo analyte ions exhibited more intense signals in the electrospray ionization positive mode than in the negative. Thus, only protonated species [M + H] ^+^ present were selected; NBPT 168 > 74.10 *m*/*z*, 168 > 94.85 *m*/*z*, 168 > 150.95 *m*/*z* and NBPTo 151.90 > 92.90 *m*/*z*, 151.90 > 134.90 *m*/*z*. Chromatograms of these transitions are shown in Figure 2. These transitions were utilised in generating the multiple reaction monitoring (MRM) method file applied for analyte detection and quantitation. The transitions with the strongest and most stable signal for NBPT (168 > 74.10 *m*/*z*) and NBPTo (151.90 > 92.90 *m*/*z*) were assigned as quantifier ions, while the qualifier ions were selected from the other transitions (NBPT, 168 > 94.85 *m*/*z*; NBPTo, 151.90 > 134.90 *m*/*z*). The quantifier product ion [M + H] ^+^ for NBPT in this study was the same as those reported in an earlier study [13]. However, the transition ions [M + H] ^+^ obtained for NBPTo in this study differed from those of Engel et al. [10], who utilised mainly the Na adduct [M + Na] ^+^ transitions. The formation of multiple adducts, including sodium adducts, are known to sometimes decrease sensitivity of the target ion [19]. To minimise or suppress adduct formation in this study and to ensure better sensitivity, acidic mobile phase additive—0.1% formic acid—was utilised. This approach was recommended by Ma and Kim [20], who also suppressed Na adduct formation species in their method using 1% acetic acid. In identifying and confirming the compounds, one precursor and two product ions (a quantifier and a qualifier ion) were used—for NBPT and NBPTo, respectively. This parameter satisfied the requirements of the European Commission for the confirmation of compounds analysed using LC-MS/MS instruments, as described in SANTE/12682/2019 guideline [18]. 

As a starting point in the chromatography method developed, different binary aqueous and organic mobile phase solvents (MeOH, H_2_O, MeCN), were assessed in various combinations. Volatile buffers (ammonium carbonate, ammonium formate or acetate) and organic acids such as formic acid and acetic acid were also evaluated as additives in the mobile phases. Their overall effects on retention time, peak shape, peak area, and baseline peak resolutions of analytes were investigated. The most satisfactory results were obtained with the combinations of mobile phase A—0.1% formic acid in 100% H_2_O and mobile phase B—90% MeCN: 10% H_2_O. The retention times (R_T_) of solvent standards of NBPTo (1.10 min), and NBPT (2.10 min), sensitivity of the method (LOQ 0.0020 mg kg^−1^), and overall run time (3.50 min) were improvements over those reported in previous studies. Van De Ligt et al. [13] noted a LOQ of 0.050 mg kg^−1^(milk samples) and total run time of 5.5 min, while Engel and colleagues [10] reported a method run time of 4 min with NBPTo and NBPT eluting at 1.35 and 2.75 min, respectively. Chromatographic separation of the analytes was achieved using a HSS T3 C_18_ column. 

### 2.3. Method Performance Characterization and Validation

#### 2.3.1. Linearity and Specificity

Eight-point procedural calibration samples, which ranged from 0.0020 to 0.250 mg kg^−1^ were injected at the beginning and end of ten analytical runs. A calibration curve was generated from the 16 points with a 1/x weighted fit and the linearity evaluated for both analytes. Linearity was assessed by visual inspection of the calibration curves, residual plots, and regression coefficient values. The coefficient of regression values (R^2^) for each of the ten analytical runs were all greater than 0.990. Accuracies of the calibration points were between 80–120% and deviations of back calculated concentrations from true concentrations were ≤ ±20%, hence satisfying the criteria for acceptance in line with SANTE/12682/2019 recommendations [18]. The specificity of a method is defined as the ability of the method to distinguish and measure the analyte of interest in the presence of structurally related compounds or drugs expected to be concomitantly administered [21]. For this reason, 30 negative samples were injected separately alongside samples fortified at 0.0020 mg kg^−1^, while method blanks and reagent blanks were also injected in each validation run. The transitions for each compound monitored showed no isobaric nor cross-talk interferences (Figure 3). The responses of the blanks and negative controls compared with the analyte standards at LOQ levels were ≤30%. Therefore, these results satisfied the conditions for sensitivity acceptability of the method (Table 1).

#### 2.3.2. Matrix Effect

The impact of matrix effects was assessed by fortifying 30 different negative raw milk samples at 0.0020 mg kg^−1^ post-extraction and comparing the response of each sample against solvent standard at the same level. The results showed there was both enhancement and suppression effects for both NBPT (ME range = 68–112%) and NBPTo (ME range = 31–117%). This is because the individual raw milk samples came from different sources and in different batches. The phenomenon of matrix components suppressing or enhancing the signal responses of target analyte ions in a bioanalysis conducted with LC-MS technique is known to occur frequently [18,22]. A key recommendation to compensate for this matrix effect is by adopting a matrix-matched calibration. In this study, a matrix-matched calibration was applied, which minimised the effects within the acceptable range of 80–120%.

#### 2.3.3. Trueness and Precision

In order to demonstrate the trueness and precision of the method under within laboratory repeatability (RSD_r_) and reproducibility (RSD_wR_), mean recoveries and relative standard deviations were calculated for all the spike levels tested in the ten analytical runs (Table 1). The results obtained showed that trueness for the high (0.0020 mg kg^−1^) and low (0.0250 mg kg^−1^) levels tested were satisfactory and met the acceptability criteria (80–120%), with NBPT values ranging from 87–118% (low level) and 92–118% (high level), while NBPTo ranged from 80–118% (low level) and 91–118 (high level). Furthermore, the method precision also showed overall satisfactory results of ≤8% (NBPT) and ≤10% (NBPTo) for both RSD_r_ and RSD_wR_. Having met the precision acceptability criteria of ≤20%, the method developed proved to be accurate and precise for the quantitation of NBPT and NBPTo residues. The criteria for determining the robustness of this method was also satisfied by the values obtained in the average recoveries and RSD_wR_, which were also within acceptable limits. Adding to the robustness of the method, a new column from a different batch, introduced during the validation runs, caused a shift in the retention times of both NBPTo (1.10 to 1.19 min) and NBPT (2.10 to 2.19 min) in milk matrix. However, the difference in the retention time shifts were < ±0.1, which is the allowable limit recommended in SANTE/12682/2019 guidelines. Furthermore, different batches of ultrafiltration tubes and mobile phase solvents, supplied by different vendors were used during analysis. The changes in brands and batches of consumables had a negligible effect on the results obtained. This further validated the robustness of the method.

#### 2.3.4. Limit of Quantitation

The limit of quantitation (reporting limit) was determined as the lowest spike level meeting the method performance criteria of trueness and precision within laboratory repeatability and reproducibility. The value corresponded to 0.0020 mg kg^−1^, which is the second calibration level in the 8-point calibration employed in this study. Trueness and precision was monitored in all the ten validation runs. The LOQ obtained in this method is much lower than that obtained in a previous study [13] (0.050 mg kg^−1^). The difference might be because both studies used LC-MS/MS instruments from different manufacturers with different method parameter settings, etc., which most likely resulted in varying levels of detection capacity. 

### 2.4. Application of Method

#### 2.4.1. Stability Assessment of Standards in Solvent and in Milk Matrix

Understanding the stability of the compounds under different conditions (storage temperature, solvent types, sample matrix, pH and storage time) is essential in ensuring accuracy of experimental results [18]. Stability studies also enable identification of optimum storage conditions for experimental samples to ensure target analytes for quantitation are not degraded before and during analysis. Limited data exist on the stability of NBPT and NBPTo in different solvents and in milk matrix, hence the importance of this study. 

The stability of 1 mg mL^−1^ NBPT and NBPTo standard solutions were assessed in four solvents (MeOH, MeCN, DMSO, and H_2_O) at different storage temperatures (+4, −20, and −30 °C) for a period of six weeks. The results showed that NBPT residues were most stable in MeOH at all temperatures. NBPTo was also stable in MeOH at all temperatures tested, except at +4 °C, where it was stable for less than six weeks (Appendix A). Consequently, standard stock solutions were prepared in MeOH and stored at −30 °C. Working standard solutions were prepared on the day they were used.

Stability assessment in milk matrix, fortified with 0.0020 and 0.0250 mg kg^−1^ of both NBPT and NBPTo standard mix solutions, indicated stability in raw and pasteurized milk samples stored at −80 °C for 4 weeks. Maintaining the milk at pH 8.5–9.0 by addition of ammonium carbonate buffer ensured stability of NBPT and NBPTo at all temperatures (−20 and −80 °C) for four weeks in raw milk samples (Figure 4). Furthermore, maintaining the milk at pH 8.5–9.0 reduced degradation of residues in vials placed in autosampler up to 24 h, and ensured repeatability of injections in the LC instrument. The acceptance criteria of ±15% difference, based on Gaugain et al. [23] criteria, was applied in determining stability. The results on stability of NBPT/NBPTo residues in milk matrix are similar to those of previous study by Van De Ligt et al. [13]. The studv also reported stability of NBPT (1 and 10 mg kg^−1^ spike) in milk matrices for 28-day storage at −80 °C, although the concentrations utilised in this study are much lower and realistic of levels expected in milk. 

#### 2.4.2. Farm Scale Pasture Grazing Experiment

This newly developed method was applied in the analysis of milk samples collected from a farm scale experiment put in place to access the potential for NBPT (applied together with N fertiliser) and the degradation product NBPTo to bio-transfer from pasture to the grazing cow milk. The residues are expected to survive the mildly acidic conditions of the rumen, be absorbed across blood-milk barrier, and be deposited in milk as contaminants. A total of 276 samples were collected from the bulk milk tank, where the milk from the cows in the Johnstown Castle experiment dairy farm was deposited. Urea + NBPT was used as the sole fertiliser N source for the red, yellow, green, and blue herds. In the case of the white herd, urea + NBPT was used to deliver c. half of the N. As a negative control, milk was also sampled from a commercial farm where no urea + NBPT was used (control farm). In addition, 240 samples were collected from 20 cows individually in Johnstown Castle. These 20 cows comprised the “green” herd and were grazing pastures fertilised at the highest rate of urea + NBPT i.e., 234 kg N/ha. Quality control samples were also analysed (Table 2). The results obtained showed that NBPT and NBPTo concentrations in the milk were below the reporting limit/limit of quantitation (0.0020 mg kg^−1^) in all samples. Given the low LOQ achieved using this method, it is expected that any residues derived in dairy products are highly unlikely to pose any toxic threat to humans. This is because NBPT is normally present at very low levels (0.038–0.064 wt.% of urea) in the final fertilizer formulations [14].

## 3. Materials and Methods

### 3.1. Chemicals and Reagents

All reagents were of analytical quality. Ultra-pure Water (Millipore 18.2 MΩcm) was generated in-house using a Milli-Q water purification system from Merck-Millipore (Cork, Ireland). Methanol 215 and Acetonitrile 200 far UV were purchased from ROMIL Ltd. (Cambridge, UK). Formic acid (98–100%) and N-(n-butyl) thiophosphoric triamide (NBPT) 98% purity were supplied by Sigma-Aldrich (Arklow, Ireland). Ammonium carbonate salt—certified for HPLC—was bought from Fisher Scientific (Blanchardstown, Ireland), while N-(n-butyl) phosphoric triamide (NBPTo) was sourced from Santa-Cruz biotechnology (Heidelberg, Germany). The chemicals and reagents were stored in line with the manufacturers’ recommendations.

### 3.2. Standard Preparations

Primary stock solutions of NBPT (molecular weight 167.21, purity 98%) and NBPTo (molecular weight 151.15, purity 98%), were prepared at concentrations of 1 mg mL^−1^ in MeOH. Aliquots of stock solutions were transferred into amber glass vials and stored at –30 °C. The choice of solvent and storage temperature was based on a preliminary in-house stability assessment.

Working standard mixed calibration solutions of NBPT and NBPTo were prepared (on the day of analysis) by diluting secondary stock standard solutions (10 µg mL^−1^ prepared same day) in methanol to give concentrations of 20, 40, 100, 200, 500, 1000, 2500, and 5000 ng mL^−1^.

### 3.3. Calibration and Quality Control

For quantitation of target analytes in milk samples, an 8-point matrix-matched calibration curve was prepared by fortifying 4 g of blank milk matrix with 200 µL of solvent standard calibrants to give final concentration values of 0.001, 0.002, 0.005, 0.010, 0.025, 0.050, 0.125, and 0.250 mg kg^−1^. Additionally, recovery control samples (blank/negative raw milk samples from bulk tanks separate from those used during the experiment) spiked post-ultrafiltration extraction at concentrations equivalent to calibration levels 2 and 5 in the milk matrix were analysed in every batch.

### 3.4. Control of pH Using a Buffer

Ammonium carbonate (1 M, pH 9.4) solution (200 µL) was added to milk to maintain a pH of 8.5–9.0 before fortification with solvent standards. This was necessary to reduce degradation of analytes during analysis.

### 3.5. Sample Preparation

Frozen milk samples were thawed by placing them in a cold water-bath for 15–20 min. The contents were then transferred into Amicon^®^ Ultra-4 Centrifugal filters (Ultracel^®^-3K) regenerated cellulose 3000 NMWL tubes (Merck, Ireland). The tubes were lightly screw-capped and allowed to stand for 10 min at 4 °C to activate the filters, before being placed in refrigerated centrifuge, Rotanta 460R (Hettich, Tuttlingen, Germany). Centrifugation was at 4000 relative centrifugal force (rcf) at 4 °C for 15–30 min. The clear filtrate of experimental samples containing NBPT and NBPTo (300 µL), and recovery samples (400 µL) spiked (with 19 µL of calibrant 2 for low level, and calibrant 5 for high level, respectively) post-extraction were placed in HPLC vials.

### 3.6. UHPLC-MS/MS Method Development and Analysis

Chromatographic separation was performed on an Acquity UPLC instrument equipped with HSS C_18_ column (2.1 × 100 mm, 1.8 µm particle size) attached to a VanGuard pre-column (2.1 × 5 mm) with similar packing material, all from Waters Corporation (Milford, MA, USA). Column temperature was maintained at 35 °C while HPLC autosampler compartment was at 10 °C. A binary gradient separation comprising of 0.1% formic acid in H_2_O (Mobile phase A), and MeCN: H_2_O (90:10, *v*/*v*—Mobile phase B), pumped at a flow rate of 0.6 mL min^−1^ was employed. Analysis was carried out using a linear gradient: 0.0–1.10 min (90% A); 1.10–1.95 min (0–50% A); 1.95–2.45 min (50% A); 2.45–2.50 (50–90% A); and 2.50–3.50 (90% A) for column clean-up and equilibration. An injection volume of 5.0 µl was used for all the samples.

The Acquity UPLC instrument, was coupled to Micromass Quattro Premier Triple Quadrupole mass spectrometer, Waters Corporation (Milford, MA, USA) fitted with an electrospray ionisation probe. Conditions were optimised by teed infusion of a 500 ng mL^−1^ standard solutions and mobile phase. NBPT and NBPTo were found to ionise in positive electrospray ionisation mode to form the pseudomolecular ion [M + H] ^+^. Following lower energy collision induced dissociation experiments, a satisfactory number of product ions were identified for NBPT and NBPTo (Table 3). The ion source parameters were as follows: capillary voltage 2.00 kV; cone voltage 20 V; inter channel delay 0.005 s; source temperature 120 °C; desolvation temperature 400 °C; cone gas flow 150 L h^−1^; desolvation gas flow 1100 L h^−1^. Instrument control and acquisition was performed with Waters MassLynx software, while data processing was done using the Waters TargetLynx software tool.

### 3.7. Method Performance Characterization and Validation

A within-laboratory method characterization and validation was performed in line with European Commission guidance documents on analytical quality control and method validation procedures for pesticide residues and analysis in food and feed [18]. Extraction method recoveries were determined by calculating average percentage recoveries from duplicate milk samples fortified at low (0.0020 mg kg^−1^) and high (0.0250 mg kg^−1^) levels. This was performed for both analytes, in each batch of ten (10) different analytical runs. Sensitivity/linearity was assessed based on an eight-point procedural calibration curve (range 0.0010 to 0.250 mg kg^−1^). The calibration samples were injected at the beginning and end of an analytical run and a curve generated from the 16 points. A matrix–induced suppression/enhancement effect study was carried out by fortifying 30 different negative milk samples at level two (0.0020 mg kg^−1^) post-extraction, and comparing the response against solvent standards at the same level. Percent matrix effect (% ME) was calculated as the ratio of the mean concentrations of the analyte (post-extracted) in the matrix and solvent standard at the same level multiplied by 100. Acceptance criteria was based on values falling within the normal range of 80–120% (Suppression, <80%; Enhancement, >120%). The limit of quantitation (LOQ) was assessed as the lowest calibration point which can be quantified while considering repeatability and precision. It was assessed by fortification of five (5) samples at the 0.0020 mg kg^−1^ in a single analytical run (repeatability) and two (2) samples at the same level in ten analytical runs (reproducibility) with trueness and precision measured in each of these analytical runs. The method performance criteria for trueness and precision was used as acceptance criteria. Trueness was determined as calculated recoveries for spike levels tested with average levels between 70–120% as acceptable. Precision, determined as relative standard deviations (RSD) within laboratory repeatability and reproducibility (RSDr and RSDwR, respectively), was assessed by fortifying five replicate samples at low (0.0020 mg kg^−1^) and high (0.0250 mg kg^−1^) concentration levels, respectively. The samples were extracted, and injected in a single analytical run with acceptance criteria taken as values that fall within ≤20% of RSD. Specificity assessment was determined by injecting 30 negative samples in the same run with samples fortified at 0.0020 mg kg^−1^, while method blanks and reagent blank were also injected in each validation run. The transitions for each compound were monitored for isobaric and cross-talk interferences. Negative samples, reagent blanks, were also injected in each validation run and the responses evaluated. Responses (*S:N*) <30% of the LOQ response were considered acceptable as basis for specificity of method. Robustness of the method was determined as the average recovery and within laboratory reproducibility (RSD_wR_) derived from the ongoing validation runs as well as assessing the extent of shift in retention times following a change in specific method parameter (column change).

### 3.8. Stability Assessment of NBPT and NBPTo in Solvents and Milk Matrix

The stability of standard materials are determined based on calculations of the percentage difference (% D) between the mean signal responses (R) of freshly prepared and old standard solutions. Different factors (uncertainties of weighing, dilution, and pipetting) can influence signal responses in LC-MS/MS analysis hence, causing old standards to give higher responses than the freshly prepared ones. To account for the effect of these factors, the equation recommended by Khedr et al. [24], which depends always on using the greater response value between the old and new solutions (*R_G_*) as the denominator, is used:(1)$$\%D=\frac{R_{new}-R_{old}}{R_{G}}\times 100$$

*R_G_* = *R_new_* when, *R_new_* > *R_old_*; or *R_G_* = *R_old_* when, *R_old_* > *R_new_*

A preliminary experiment was undertaken to determine the most appropriate solvent and temperatures for preparation and storage of solvent standards and samples for routine analysis. The experimental design proposed by Berendsen et al., [25] was adopted with some modifications.

For solvent standard stability assessment, mixed stock solutions of NBPT/NBPTo, 1.00 mg mL^−1^, was prepared in replicates of 3 in MeOH, MeCN, DMSO, and H_2_O solvents, respectively. These were stored at +4, −20, or −30 °C for 6 weeks in duplicates. A fresh set of standards was prepared each week for week 6 and compared with freshly prepared standards at the end of six weeks storage.

In milk matrix, the stability of NBPT and NBPTo were assessed in raw and pasteurised milk samples (4 ± 0.04 g), fortified in replicates of 3 at concentrations of 0.0020 and 0.0250 mg kg^−1^, respectively. Stability was investigated at two different temperatures (−20 and −80 °C) for storage times up to 4 weeks. In addition, a set of these stability samples were adjusted to pH 8.5–9.0 using ammonium carbonate buffer (1 M, pH 9.4) to prevent analyte degradation.

At the end of the experiments, the samples were processed and analytes extracted, before analysis was performed, using the UHPLC-MS/MS method developed. Five replicate injections were made for each sample set. Means and relative deviations of replicate injection responses were determined and compared with those of newly prepared standards at the same level. As NBPT rapidly converts to NBPTo in solution, the sum of the residues (NBPT and NBPTo) were determined in the milk matrix and utilised in the calculation of percentage differences.
(2)$$NBPT_{Sum\;of\;Residues}=\left(1.00\times C_{NBPT}\right)+\left(1.10\times C_{NBPTo\;}\right)$$
where $C_{NBPT}$ and $C_{NBPTo\;}$ are calculated concentrations of NBPT and NBPTo residues and the constants 1.00 and 1.10 are conversion factors of NBPT and NBPTo, respectively.

An acceptance limit of ±15% in the percentage difference was applied in this study. Although European Commission guidelines [18] recommends ±10%, Gaugain et al. [23] and other studies [25,26] have reported that this acceptance limit (±10%) might be less than the variability of analytes responses due to inherent variability of response associated with LC-MS/MS techniques. Therefore, a higher limit of ±15% may be applied based on the detectable difference of the means by the analytical method applied, while putting into account repeatability parameters.

### 3.9. Farm Scale Pasture Grazing Experiment

The experiment was conducted at the Teagasc, Johnstown Castle Research Centre, Co., Wexford, Ireland (52°18′ N, 6°30′ W). The well-being of animals and the protocols used in this study are in accordance with the European Council Directive (98/58/EC) [23].

Five herds of Holstein-Freisian dairy cows (*n* = 155) were grazed on perennial ryegrass (*Lolium perenne* L.) and white clover (*Trifolium repens* L.) swards or a multi-species sward (perennial ryegrass, white clover, red clover (*Trifolium pratense*), timothy (*Phleum pratense* L.), chicory (*Cichorium intybus* L.), and plantain (*Plantago lanceolata* L.) The swards were managed in a rotational grazing system. A urea + NBPT nitrogen fertiliser was blanket spread on the grazing platform in split applications to correspond with the grazing rotation duration over the course of the growing season. The total annual rates for each herd are stated in Table 4.

Milk was collected at the dairy parlour from two point sources: (i) the bulk tank and (ii) the individual cows. Raw, unpasteurised milk (4 °C) was collected from the bulk tank once in the morning (AM) and evening (PM) for 54 weeks from 26 February 2020 to 11 February 2021. A control herd on a commercial farm was selected where urea + NBPT was applied. Milk was sampled from these lactating cows, for 28 weeks, between March and October 2020. The samples were stored at ≤4 °C during transportation from the farm to the laboratory at Johnstown Castle.

Milk was also collected from individual cows (*n* = 20) in the green herd, which received the highest N rate of 234 kg N ha^−1^ and thus the highest NBPT loading every two weeks between 15 June and 24 August 2020. Following collection, both the bulk tank (*n* = 3) and individual cow (*n* = 20) milk samples were transferred into 4 mL tubes (Sarstedt, Nümbrecht, Germany), adjusted to pH 8.5–9.0 using ammonium carbonate buffer (1 M, pH 9.4) and stored at −80 °C. Frozen samples were shipped in dry ice for analysis.

## 4. Conclusions

A robust and sensitive analytical method, based on a one-step ultrafiltration extraction technique, and followed by UPLC-MS/MS detection, was developed for quantitative analysis of NBPT and NBPTo in a milk matrix. The method has been validated extensively, over two concentration levels based on literature review and in line with expected concentrations in the target matrix. The method is more sensitive and less laborious in comparison with other existing methods. This was made possible by the use of ultrafiltration tubes that involved only transfer of samples into the tube, followed by ultra-centrifugation. Furthermore, addition of ammonium carbonate buffer enhanced the stability of the compounds in the milk matrix at room temperature and in the autosampler during injection, thereby stabilising residue compounds for ionisation and subsequent detection by the instrument. This study also highlights the importance of using matrix-matched calibration, which reduced the matrix effect during analysis. The method was validated in line with approved guidelines published by the European Commission and is considered fit-for-purpose for the intended purpose. Analysis of 516 milk samples from the farm study, found that NBPT and NBPTo concentrations were below 0.0020 mg kg^−^^1^ (the reporting limit/limit of quantitation achieved by the method developed). Thus, it is expected that any residues derived in dairy products are highly unlikely to pose any toxic threat to humans since NBPT is normally present in very low levels (0.038–0.064 wt.% of urea) in the final fertilizer formulations [14].

## Figures and Tables

**Figure 1 molecules-26-02890-f001:**
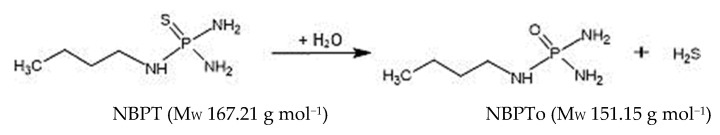
Conversion of NBPT to NBPTo.

**Figure 2 molecules-26-02890-f002:**
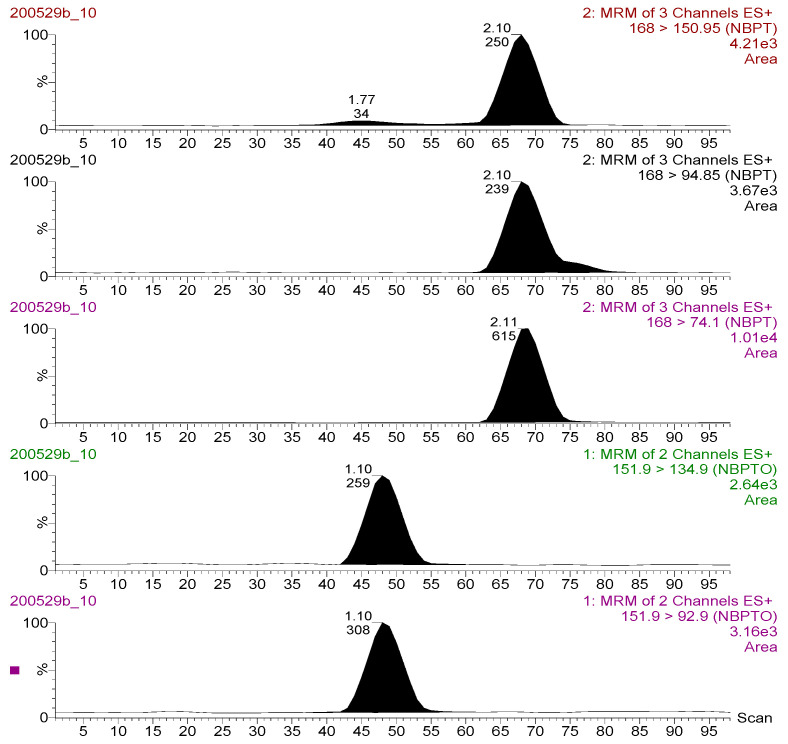
Chromatograms for the transition ions monitored for NBPT (upper three: 168 > 150.95; 168 > 94.85; 168 > 74.10 *m*/*z*) and NBPTo (lower two: 151.90 > 134.90; 151.90 > 92.90 *m*/*z*) residues (standard solution 500 ng mL^−1^) analysed in LC-MS/MS ESI+ mode.

**Figure 3 molecules-26-02890-f003:**
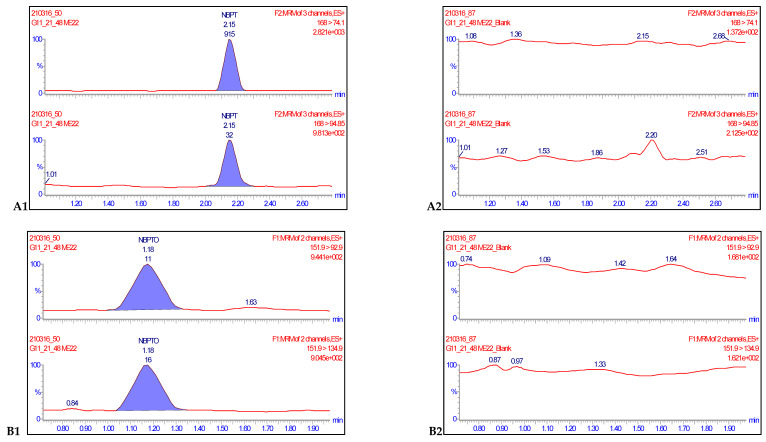
Representative chromatograms of NBPT (**A1**) and NBPTo (**B1**) quantifier and qualifier ion transitions spiked in negative milk samples at concentrations of 0.0020 mg kg^−1^ and their corresponding blanks NBPT (**A2**) and NBPTo (**B2**) showing no isobaric or cross-talk interferences.

**Figure 4 molecules-26-02890-f004:**
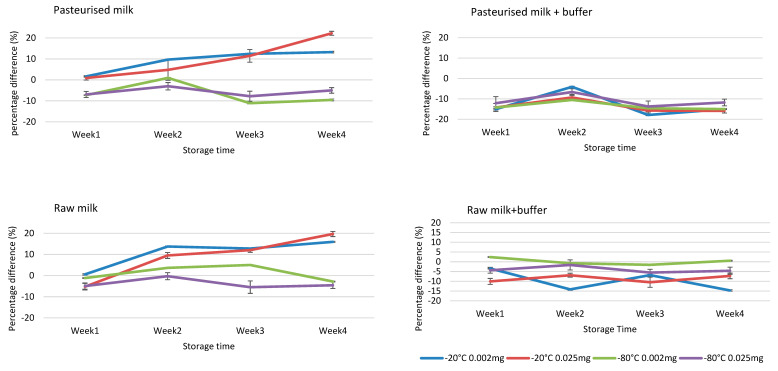
Stability assessment of NBPT and NBPTo in raw and pasteurised milk samples at concentrations 0.0020 and 0.0250 mg kg^−1^, storage temperatures −20 and −80 °C for 4 weeks. The residues were stable for four weeks in samples spiked at both 0.0020 and 0.0250 mg kg^−1^, stored at −80 °C with or without buffer treatment. The addition of ammonium carbonate buffer in raw milk stabilised the residues at both storage temperatures for four weeks. Stability acceptance criteria = percentage difference of ±15% [23]. Concentration values obtained were expressed as sum of residues (NBPT_Sum of residues_) ± SD using Equation (2), while percentage difference was calculated using Equation (1).

**Table 1 molecules-26-02890-t001:** Summary of validation data.

Validation Parameter (10 Runs)	NBPT		NBPTo	
Level (mg kg^−^^1^)	L1, 0.0020	L2, 0.0250	L1, 0.0020	L2, 0.0250
Extraction recovery				
Average Accuracy (%)	92	95	94	98
Precision (%)				
RSDr	6	1	9	3
RSDWR	8	7	10	6
Trueness (%)				
Within laboratory repeatability, WLr	103	101	100	108
Within laboratory reproducibility, WLR	102	99	101	103
Sensitivity/Linearity				
Accuracy Range (%)	81–119		81–117	
RSD (%)	6		8	
R2	≥0.9940		≥0.9900	
Reporting Limit/ LOQ (mg kg^−^^1^)	0.0020		0.0020	
Matrix Effect range (*n* = 30), %	68–112		31–117	
RSD (%)	13		16	

**Table 2 molecules-26-02890-t002:** Sampling periods and summary result of NBPT and NBPTo residues in milk sampled on a farm using NBPT treated urea (NBPT Farm) and a farm not using NBPT treated urea (control farm).

Sampling Year	Month	Sample Source		
		Bulk Tank		Individual Cows
		Experiment Farm Samples (Johnstown Castle Farm)	Negative Control Farm (No urea + NBPT)	Grazing Pastures Fertilised at 234 kg N/ha as Urea + NBPT
2020	February	12	-	-
	March	24	3	-
	April	24	9	-
	May	24	12	-
	June	24	9	80
	July	24	9	80
	August	21	9	80
	September	26	8	-
	October	10		
	November	8		
	December	8		
2021	January	8		
	February	4		
Total sample		217	59	240
Sum of residues (NBPT and NBPTo) concentrations for all samples (mg kg^−^^1^)		<0.0020	<0.0020	<0.0020

**Table 3 molecules-26-02890-t003:** Mass Spectrometer settings for MRM of 5 mass pair in ESI+ mode.

Channel	Compound	RT (min)	Parent ion (*m*/*z*)	Daughter (*m*/*z*)	Dwell Time (s)	Collision Energy (eV)
1	NBPTO	1.10	151.90	92.90	0.200	15.00
2			151.90	134.90	0.200	13.00
3	NBPT	2.11	168.00	74.10	0.200	12.00
4			168.00	94.85	0.200	18.00
5			168.00	150.95	0.200	10.00

**Table 4 molecules-26-02890-t004:** Description of the individual dairy cow herds and paddocks.

Herd	Number of Cows	Nitrogen Rate (kg N ha^−1^)	Sward
Red	44	214	Ryegrass/Clover (95%/5%)
Yellow	42	212	Ryegrass/Clover (95%/5%)
Green	20	234	Ryegrass/Clover (80%/20%)
Blue	20	82	Multi-species
White *	29	162	Ryegrass/Clover (95%/5%)
**Mean N Rate**		**181**	

* Protected urea was spread on 50% of the grazed paddock area.

## Data Availability

Data is available on request from authors.

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
