# Peer review of "Development of One-Step Non-Solvent Extraction and Sensitive UHPLC-MS/MS Method for Assessment of N-(n-Butyl) Thiophosphoric Triamide (NBPT) and N-(n-Butyl) Phosphoric Triamide (NBPTo) in Milk"

_molecules, 2021, doi:10.3390/molecules26102890_

Round 1
Reviewer 1 Report
The work is well done, and many samples have been analyzed by using the developed strategy. I suggest some minor revisions.
Please specify in the title the meaning of NBPT and NBPTo
Please indicate in figure 1 what is lost (-H2S), the mass value of NBPT and NBPTo, the structures with the same geometry.
Line 97: change “was developed” in “was applied”
Line 115: correct the reference style (Lee et al. 2003)
Line 132: please add error to 93% recovery
Line 142: please rewrite the sentence “NBPT 168 > (74.10, 94.85, 150.95 m/z) and NBPTo 151.90 > (92.90, 134.90 m/z)” clarifying the meaning of molecular ion and associated fragments.
How many replicates have been used for the stability experiments reported in figure 4? Please specify and add a bar error to figure 4.
I cannot see supplementary material. I suggest adding the structures of qualifier and quantifier fragments in supplementary files. More, I suggest describing which kind of degraded products can be generated at acidic pH.
Please check the very recently published papers on the topic.
Cite: Cantarella et al. Agronomic efficiency of NBPT as a urease inhibitor: A review Journal of Advanced Research, 2018, 13, 19-27
Janke et al. Geoderma 2021, 382,114770
Mateo-Marín, N., Quílez, D., Isla, R., Journal of Plant Nutrition and Soil Science, 2020, 183, 567
Reviewer 2 Report
The presented work explains the validation of the uHPLC-MS method of NBPT and NBPTo, all the validation parameters and their potential use in the measurement of these components in milk are observed. A quick ultra-filtration extraction and ultra-high performance liquid chromatography coupled to mass spectrometry triple quadrupole (UHPLC-MS/MS) quantitation method, developed and validated in this study. The method is deemed fit-for-purpose for the simultaneous determination of NBPT and NBPTo in milk.
Development of one-step non-solvent extraction and sensitive UHPLC-MS/MS method for assessment of NBPT and NBPTo in milk
Abstract:
Some validation parameters were missing to be mentioned, such as linearity (coefficient of determination or correlation, response factor); explain the mean standard deviation between and within measurement.Explain the two levels for precision and accuracy.
Introduction
The introduction is adequate, it puts in context in an adequate way what is the problem of the toxic compound, the adequate motivation to generate a method sensitive to low concentrations for its determination.
Line 46. Reference
Line 51. Reference
Line 76 - 89. Reference
Figure 1. Poor quality.
Figure 4. Poor quality.
Results and Discussion
Sample Extraction development
The methodology used remains to be discussed further, they only present a study, it is not enough for a discussion.
UHPLC-MS/MS method development
Some elements are missing in the discussion of the standardization of the use of uHPLC-MS, complete.
Trueness and Precision
It does not present discussion or comparison with other similar methods.
Limit of Quantitation
It does not present discussion or comparison with other similar methods.
Does not mention Calibration and quality control
Line 480 to 482. nitrogen fertiliser was blanketspread on the grazing platform in split applications to correspond with the grazing rotation duration over the course of the growing season. Text blue.
Conclusion
Not comment
